# On the Neurobiology of Meditation: Comparison of Three Organizing Strategies to Investigate Brain Patterns during Meditation Practice

**DOI:** 10.3390/medicina56120712

**Published:** 2020-12-18

**Authors:** Frederick Travis

**Affiliations:** Center for Brain, Consciousness and Cognition, Maharishi International University, Fairfield, IA 52557, USA; ftravis@miu.edu; Tel.: +1-641-472-1209

**Keywords:** meditation, focused attention, open monitoring, Transcendental Meditation

## Abstract

Three broad organizing strategies have been used to study meditation practices: (1) consider meditation practices as using similar processes and so combine neural images across a wide range of practices to identify the common underlying brain patterns of meditation practice, (2) consider meditation practices as unique and so investigate individual practices, or (3) consider meditation practices as fitting into larger categories and explore brain patterns within and between categories. The first organizing strategy combines meditation practices defined as deep concentration, attention to external and internal stimuli, and letting go of thoughts. Brain patterns of different procedures would all contribute to the final averages, which may not be representative of any practice. The second organizing strategy generates a multitude of brain patterns as each practice is studied individually. The rich detail of individual differences within each practice makes it difficult to identify reliable patterns between practices. The third organizing principle has been applied in three ways: (1) grouping meditations by their origin—Indian or Buddhist practices, (2) grouping meditations by the procedures of each practice, or (3) grouping meditations by brain wave frequencies reported during each practice. Grouping meditations by their origin mixes practices whose procedures include concentration, mindfulness, or effortless awareness, again resulting in a confounded pattern. Grouping meditations by their described procedures yields defining neural imaging patterns within each category, and clear differences between categories. Grouping meditations by the EEG frequencies associated with their procedures yields an objective system to group meditations and allows practices to “move” into different categories as subjects’ meditation experiences change over time, which would be associated with different brain patterns. Exploring meditations within theoretically meaningful categories appears to yield the most reliable picture of meditation practices.

## 1. Introduction

Meditation practices have become part of the business, education and self-help cultures in the West [1]. A search on Pubmed with “meditation” as the key word yielded 7340 hits. To understand the nature of meditation practices, different strategies have been used to group individual studies into larger categories for analysis. This paper compares three strategies to organize meditation research. These three strategies are: (1) consider meditation as using similar cognitive process and so research the common underlying brain patterns of meditation, (2) consider each meditation practice as unique and so investigate individual practices and report patterns of each practice, or (3) consider that meditations fit into larger categories and compare brain patterns within and between categories.

This paper is not an exhaustive review of all papers on meditation practices. Rather, it compares the conclusions from these three strategies to group the data. It is written to alert researchers to the impact of different organizing strategies on their conclusions from meditation research.

## 2. First Strategy: Identify Underlying Brain Patterns of Meditation Practices

If a common brain pattern underlies most meditation practices, then combining many trials through principal components analysis or averaging should bring out the assumed underlying pattern. This strategy has been used in psychological and EEG (electroencephalography) research. In psychometric studies of intelligence, the first unrotated factor in a principal components analysis of intelligence tests is considered to indicate a broad mental capacity that influences performance across a range of reasoning and problem solving tests, called “g” or general intelligence [2,3]. In the EEG domain, EEG signals from successive stimulus presentations are time locked to the stimulus onset and then averaged. The averaging reduces the noise and brings out, through constructive interference, the brain components common in that task [4]. In each of these examples, a common, underlying pattern emerges from the analysis.

Table 1 lists a sampling of studies that used the “underlying-pattern” organizing principle. Notice the range of meditation practices included in each study and the lack of convergence of the results they report. The studies are listed chronologically.

### 2.1. Changes in Gray Matter Volumes

Luders and colleagues reported no group differences in total cortical thickness, but higher grey matter volumes in right sub-cortical areas in the meditation group [5]. In contrast, their 2019 research reported higher grey matter volume and thicker cortices overall in the meditation group [12]. These contradictory findings could reflect different contributions of different practices in each meditation study.

Fox and colleagues reviewed 21 neuroimaging studies and found thicker cortices in brain areas supporting a wide range of cognitive processes including meta-awareness (frontopolar cortex), body awareness (sensory cortices and insula), memory consolidation and reconsolidation (hippocampus), self and emotion regulation (anterior and mid cingulate; orbitofrontal cortex) and intra- and interhemispheric communication [8]. This wide range of meditation effects could reflect the contributions of different meditation practices. Unexpectedly, Fox and colleagues reported a *negative* correlation with magnitude of brain differences and length of meditation practice. This suggests that other factors besides meditation practice may be influencing these findings.

Grey matter volume decreased with age with all subjects, though the decrease was less in the meditating subjects [7]. At age 50, brains of meditators were estimated to be 7.5 years younger than those of controls [11]. This finding could be a characteristic of most meditation practices since experience is reported to affect grey matter volumes across the lifespan [14]. In addition, yoga practices with and without meditation and with and without postures reported higher insula and hippocampal grey matter volumes.

### 2.2. Changes in White Matter Volumes

Two studies reported that fractional anisotropy, an indicator of white matter fiber integrity, was higher in the meditating population. One was a study of 27 meditating and control subjects [6]. The second was a meta-analysis of nine studies [12]. White matter changes may also be common to all meditations, different meditation procedures would be expected to affect different brain networks since different kinds of experiences change white matter networks in different ways. For instance, studying for the Law School Administration Test increased white matter tract integrity in frontal executive areas [15], while juggling increases white matter integrity in the right posterior parietal cortex, related to detecting motion [16].

### 2.3. Summary of this Section

The assumption that a common brain marker would emerge by combining different meditation practices together in one analysis is flawed. Meditation procedures differ. Some meditations involve deep concentration, others prescribe attention to external and internal stimuli [17], and others are inwardly directed towards nondual states [18]. Thus, brain patterns from different practices would not be expected to converge to a common pattern. Looking at brain areas reported active during meditation practices in the above table, we find all brain areas—except for primary sensory cortices—were reported to be activated by meditation practices.

## 3. Second Strategy: Researching Individual Meditation Practices

Research has investigated shorter term (state) and long term (trait) brain patterns during individual meditation practices.

Buddhist Shamatha meditation, a concentrative technique, is associated with activation in the fronto-parietal attention network and deactivation of regions related to conceptual thought and emotions [19,20].Loving-kindness/compassion meditation in expert meditators activated limbic areas and a network associated with Theory of Mind—right temporal lobes, temporo-parietal junction, medial prefrontal and posterior cingulate cortices [21,22,23].A meta-analysis of 16 fMRI studies of compassion meditation revealed patterns of activation in four brain areas: periaqueductal grey, anterior insula, anterior cingulate, and inferior frontal gyrus. This meta-analysis replicated earlier findings but did not find activation in areas typically connected with compassion including the dorsolateral prefrontal cortex, orbital prefrontal cortex and the amygdala [24].Mindfulness meditation was marked by deactivation of medial prefrontal cortices but activation of lateral prefrontal cortices, the secondary somatosensory cortex, the inferior parietal lobule and the insula [25,26,27,28].A meta-analysis of 21 fMRI studies of mindful practices used activation likelihood estimate methods to compare mindfulness with control conditions. Loci of activation were found in the frontal regions including the medial prefrontal gyrus, anterior cingulate, insula, and globus pallidus [29].Vipassana expert meditators had bilateral activation in the rostral anterior cingulate cortex and in the dorsal medial prefrontal cortex [30,31,32].Zen meditation is reported to lead to increased alpha and theta activity in many brain regions, including the frontal cortex, and decreases in the Default Mode Network [33,34,35].Network analysis of imaging data from 12 experienced Zen meditators and 12 controls during an attention to breathing protocol reported extensive connections of frontoparietal circuits with early visual and executive control areas [36].Relaxation meditation (Yoga Nidra) activated the hippocampus and posterior mental imagery areas, and deactivated the frontal executive control network system, in a positron emission tomography study [37,38].Kundalini yoga activated left fronto-temporal regions and deactivated the left posterior parietal lobe in a PET study [39,40].Mantra recitation with meaning activated the hippocampi/parahippocampal area, middle cingulate cortex and the precentral cortex bilaterally [41].Kirtan Kriya activated the medial prefrontal cortex and left caudate nuclei and deactivated the left superior occipital and inferior parietal cortex, as well as the right inferior occipital cortex [42,43].Ashtanga yoga deactivated the medial prefrontal cortex, anterior cingulate gyrus, and inferior parietal cortices [42,44].Transcendental Meditation practice is marked by frontal alpha1 coherence (8–10 Hz) [45,46,47], and ventral medial and anterior cingulate sources of alpha activity, in an magnetoencephalography study [48], and higher frontal and lower brainstem blood flow in an fMRI study [49].

### Summary of this Section

This overview demonstrates that meditation practices have pervasive effects on brain functioning, activating and deactivating different brain networks. For instance, mindfulness, Vipassana and TM activate the dorsolateral prefrontal cortex, but only TM also activates the medial prefrontal cortex, part of the emotional regulation network in the brain [38]. Loving kindness and Kirtan Yoga also activate the medial prefrontal cortex but do not affect dorsolateral prefrontal cortex functioning. Mindfulness practices activate the inferior parietal cortices, as does Yoga Nidra, but Yoga Nidra deactivates dorsolateral prefrontal cortex. 

This organizing strategy generates a multitude of brain patterns as each practice is studied individually. Research should be done to understand the details of each practice. However, the rich detail of individual differences within each practice makes it difficult to identify reliable patterns between practices.

## 4. Third Strategy: Organizing Meditations into Larger Classes

A third organizing strategy is to group practices based on theoretical frameworks and identify neural patterns within and between each group. This should reduce the variability within and between groups and so lead to a more reliable picture of brain patterns during meditation. We will review the impact of three grouping strategies: grouping meditations by their traditions—Indian or Buddhism traditions, grouping meditations by their procedures, and grouping meditations by EEG frequencies associated with the cognitive processes used in each practice.

### 4.1. Grouping Meditations by Their Source: Indian or Buddhist Traditions

Most eastern meditation practices were developed within the Indian and Buddhist traditions. Meditations in the Buddhist tradition are dynamic cognitive practices [50]. They include mindfulness as a central aspect, i.e., sustained attention and mental presence on breathing, on the contemplation of the body and on the observation of the contents of the mind [51]. During mindfulness practices one attends to “…whatever predominates in awareness, moment to moment. The intention is not to choose a single object of focus, but rather to explore changing experience” [52]. This cognitive process—attending nonjudgmentally to the unfolding of experience moment by moment—has been applied to eating [53], walking [54], awareness of breathing [55] and awareness of thoughts and emotions [56].

In contrast, meditations in the Indian tradition are structured to reach a state of content-free awareness—the state of Yoga or Samadhi [57]. The procedures in these meditations involve less cognitive control and recommend: “We simply need to be what we are.” ([58] or “Step out of the process of experiencing and arrive at the state of Being.” [59].

Chiesa and colleagues conducted a meta-analysis of eighteen neural imaging studies: ten Buddhist meditations and eight Indian meditations [57]. The practices from the Buddhist tradition included Vipassana, Zen, Loving-kindness/compassion, and Mindfulness-Based Stress Reduction. The practices from the Indian tradition included Yoga Nidra (guided relaxation), Kundalini yoga (chanting, singing, breathing exercises, and repetitive poses to move kundalini energy up the spine), chanting of ‘‘OM”, Kirtan Kriya (singing of mantras) and Shabdha Kriya Yoga (singing of mantras and breath control used before going to sleep). 

Comparing meditations in these two groups, Buddhist meditations were characterized by bilateral activation in the frontal superior medial gyrus, activation in the right parietal supramarginal gyrus (associated with language processing) and the supplementary motor area. Indian meditations were associated with left lateralized activation of the post-central gyrus (touch), the superior parietal lobe (touch), the hippocampus (memory and spatial planning), the left superior temporal gyrus (phonological word processing), and the right middle cingulate cortex (attention) [57].

The Buddhist meditations in this study enhanced brain areas associated with language processing and motor areas. Indian meditations in this study enhanced brain areas associated with sensory processing, language processing, attention, and memory areas.

Buddhist and Indian traditions include practices that involve minimal cognitive control such as Dzogchen meditation in the Tibetan Buddhist tradition, Zazen in the Buddhist tradition and Transcendental Meditation in the Indian tradition. These were not included in this meta-analysis. Dzogchen describes meditation as “…the state of relaxation—a means by which we can be what we are, without tension, tyranny or anxiety….We simply need to be what we are.” ([58], p. 14). Zazen is described as “the practice of being what we are, of allowing, permitting, opening ourselves to ourselves. In doing that we enter directly the depth of our living—a depth that goes beyond our individual life and touches all lives.” ([60], p. 26). Transcending during Transcendental Meditation practice is described as “When we have transcended the field of the experience of the subtlest object, the experiencer is left by himself … the experiencer steps out of the process of experiencing and arrives at the state of Being. The mind is then found in the state of Being.” ([59], p. 29).

Combining cognitively active meditations involving sustained attention, mental presence on breathing, contemplation of the body or observation of the contents of the mind with meditation described as only needing “to be what we are” is mixing meditations with very different procedures. This could lead to distorted conclusions about meditation practices.

### 4.2. Grouping Meditations by their Procedures: Four Categories of Meditation

Fox and colleagues started with the assumption that meditation practices with different procedures would be associated with different neural imaging patterns [17]. They defined four categories of meditation based on their procedures: *Focused Attention, Mantra Recitation, Open Monitoring, and Loving Kindness. Focused Attention* is directing attention to one specific object while monitoring and disengaging from extraneous thoughts or stimuli; *Mantra Recitation* is repetition of a sound with the goals of calming the mind and avoiding mind-wandering; *Open-Monitoring* is bringing attention to the present moment and impartially observing all mental contents as they naturally arise and subside; and *Loving-kindness* is deepening feelings of sympathetic joy for all living beings, as well as promoting altruistic [56,61].

The authors reviewed 78 functional neuroimaging (fMRI and PET) meditation studies involving 527 participants and placed the studies into these four categories [17]. In this meta-analysis, *Focused Attention* meditations included Tibetan Buddhism, Vipassana, Theravada Buddhism and Zen, *Mantra Recitation Meditation* included kundalini, ACEM, SOHAM and Pure Land Buddhist meditations, *Open Monitoring* included Yoga Nidra and Mindfulness practices, and *Loving kindness* included Tibetan Buddhist meditations.

The meditations within each category had similar activation patterns, but there was little convergence between the categories. During *Focused Attention* meditations, activations were observed in two brain areas associated with voluntary regulation of thought and action, the premotor cortex and dorsal anterior cingulate cortex. During *Mantra Recitation* meditations, areas of activations included the posterior dorsolateral prefrontal cortex, motor cortex, and putamen/lateral globus pallidus. During *Open Monitoring* meditations, significant clusters of activation were observed: the insula, left inferior frontal and motor cortices. During *Loving kindness* meditations, higher brain activation as observed in the right anterior insula/frontal operculum, and anterior inferior parietal lobule. 

These four categories of meditation differed on the level of description of their procedures and on the level of brain activation patterns [17]. This supports the value of assigning meditations to meaningful categories before combining neural imaging data.

The categories could have been further refined. *Focused Attention* and *Open Monitoring* refer to the procedures used during meditation practice. *Mantra recitation meditation* and *Loving-kindness/compassion* refer to the content of the meditation. These are distinctly different groupings—"procedures used” versus “content of the meditation.” The authors discussed this contradiction. In addition, they noted that *Mantra Recitation Meditation* could reasonably be placed within the *Focused Attention* category since most mantra meditations involve directing attention to the mantra while disengaging from extraneous thoughts [17]. Similarly, *Loving-kindness/Compassion* meditations could be considered a form of *Focused Attention* since one focuses intensively on the target of loving-kindness, while cultivating a consistent emotional tone to the exclusion of other [17].

### 4.3. Grouping Meditations by EEG Activation Patterns: Three Categories of Meditation

Travis and Shear started with descriptions of meditation procedures, as Fox and colleagues did, and then assigned frequency bands to the cognitive processes reflected in those descriptions [62]. Then, based on these EEG frequency bands, meditations were grouped into categories. These categories are procedural categories as defined by EEG frequencies and are not limited by the domain of the meditation practice, such as perceptual, cognitive, affective, or nondual domains [50].

Two categories of meditation procedures, *Focused Attention* and *Open Monitoring*, had been earlier defined by Lutz and colleagues [61] and were used by Fox. These two categories were the starting point for this three-category classification of meditations.

#### 4.3.1. Focused Attention

*Focused Attention* meditation “…entails voluntary focusing attention on a chosen object in a sustained fashion. “([61], p. 1). Voluntary focused attention is associated with gamma activity (30–50 Hz). Gamma is seen in any task that involves effortful thinking and control of the mind [63]. Gamma EEG is driven by local processing within short-range connections responsible for perceptual awareness [64], selective attention [65], neuronal processing and communication between regions [66]. In a study of experienced Buddhist practitioners, higher gamma EEG was seen in the posterior cingulate cortex, a major posterior hub in the default mode network [67].

#### 4.3.2. Open monitoring

*Open monitoring* meditations “…involve nonreactive monitoring of the content of experience from moment to moment, primarily as a means to recognize the nature of emotional and cognitive patterns” ([61], p. 1). Nonreactive monitoring is associated with midline frontal theta. Theta is produced whenever one attends to internal mental processing [32]. Frontal midline theta is a marker of working memory tasks, and episodic memory encoding and retrieval processes [68]. Brandmeyer and Delorme compared EEG correlates of novice mindfulness subjects, whose minds wandered frequently during the practice to EEG from experienced mindfulness subjects whose minds wandered less. They found higher midline theta EEG and somatosensory alpha in the experienced subjects [69]. A recent meta-analyses of EEG patterns [9] also reported that alpha was found in posterior, parietal or central areas during open monitoring practices [46].

#### 4.3.3. Automatic Self-Transcending

Travis and Shear added a third category, *Automatic Self-Transcending,* for meditations that transcend their steps of practice [62]. This would include meditation practices such as Dzogchen “We simply need to be what we are.” ([58], p. 14), and Transcendental Meditation practice “…the experiencer steps out of the process of experiencing and arrives at the state of Being. The mind is then found in the state of Being.” ([59] p. 29).

Frontal alpha EEG is predicted to characterize meditations with these procedures. Alpha oscillations have been found to play an active role in the suppression of task-irrelevant processing [66], being negatively correlated with local cortical excitability [70]. Frontal alpha could indicate that frontal executive processing is being inhibited.

#### 4.3.4. Application of this Model

Travis and Shear used these EEG frequency bands to assign meditations to categories [62]. Meditations that were characterized by gamma EEG were placed in the *Focused Attention* category. They included Compassion [61], Qigong [71], Zen [35,60] and Vipassana meditations [32,72]. The essence-of-mind technique, a Tibetan Buddhist technique aimed at experiencing a brilliantly awake, limitless, a non-dual state of awareness, is also characterized by higher gamma and beta-2 EEG [73] and would be placed in this category.

Meditations that were characterized by midline theta and posterior alpha EEG were placed in the *Open Monitoring* category. They included Mindfulness practices [28,69], Zazen [74], and Sahaja [75,76].

Meditations that were characterized by frontal alpha1 EEG were placed in the *Automatic Self-Transcending* category. They included Transcendental Meditation [45,47], a case study of a Qigong master [77] and a study of 25 Zen-Buddhist priests [78].

##### Automaticity: Why Zen Buddhism and Qigong Might Be in This Category

Gamma EEG is most often reported during Zen Buddhism and Qigong practice [79]. This is understandable since these practices require focus and control of mental processes. However, even highly complex, controlled cognitive processes can become automatic with extensive practice—the task is performed without attention to each step and performance is not affected by increasing task loads [80]. Experienced Vipassana practitioners reported “effortless doing,” a calm, tranquil, relaxed, and effortless way of acting [81]. Expert Buddhist practitioners report “effortless concentration” characterized by reduced need to consciously direct, sustain and orient attention [61]. These are examples of automaticity resulting from long-term practice.

The TM technique is designed to be automatic. One learns how to use the “natural tendency” of the mind to transcend [82]. The movement of the mind to more interesting experiences is what is intended by the phrase the “natural tendency” of the mind. When the object of experience is rated as being highly interesting, it is associated with greater positive emotional tone than when the person is on-task [83]. The mind transcends because one’s attention is pulled by the inherent pleasure of the state of inner silence devoid of changing thoughts, feelings or perceptions [49,84]. This claim is supported by high activation levels in the default mode network during Transcendental Meditation practice.

##### Default Mode Network Activation—A Marker of Automaticity 

Activation/deactivation patterns of the default mode network (DMN) are an objective marker of cognitive control used in a task [84,85]. The DMN includes medial and lateral parietal cortices, and the precuneus and medial prefrontal cortex [86,87]. The DMN is deactivated during goal-directed behaviors requiring executive control and activated during self-referential mental activity, when envisioning the future and when envisioning the actions of others [88].

All meditations in the *Focused Attention* and *Open Monitoring* categories are reported to lead to decreased activity of the default mode network [89,90] including Zen [35], Vipassana [91], mindfulness practices [92,93], and Loving-Kindness [94]. Decrease in default activation is expected since these meditation practices involve voluntary attention to a specific object or attending to moment-by-moment changing experiences. In contrast, default mode network activation remains high during Transcendental Meditation practice [84].

##### Default Mode Network and Mind Wandering

High DMN activity has also been reported during rumination and is associated with depression. Zhou and colleagues conducted a meta-analysis of 14 fMRI studies with 286 participants. High depression was marked by activation of four sub-systems of the default network: the anterior prefrontal cortex, dorsal medial prefrontal cortex, posterior cingulate cortex and posterior inferior parietal lobe [95]. This finding can be interpreted as high DMN activation causing or indicating rumination and depression. This conclusion is an example of the logical fallacy of affirming the consequent. Namely: if depression is associated with high DMN activity, then high DMN activity indicates depression in all populations.

Mind-wandering often involves negative content such as guilty daydreaming, ruminating thoughts and unpleasant emotions [96,97]. However, mind-wandering more often involves positive content rather than negative content [98]. This positive content includes future-focused, autobiographical planning [99] oriented toward personal goal resolution [100]. Positive mind-wandering allows attentional cycling between personally meaningful values and external goals [101].

Previous studies reporting a relationship between mind-wandering and depression combined positive mind-wandering, rumination, and worry under the single term “repetitive thinking” [102,103]. The negative effects of “mind-wandering” vanished when “negative” mind-wandering (perseverative cognition) is differentiated from “positive” mind-wandering. Research reports that *positive* mind-wandering and depression symptoms are not correlated at baseline or at one-year [104]. When compared to positive mind-wandering, perseverative cognition was associated with higher levels of cognitive inflexibility (slower reaction times, higher levels of intrusiveness), autonomic rigidity (lower heart rate variability), and worsening mood (more symptoms of depression) compared to positive mind-wandering [105]. (See this book chapter for a discussion of negative and positive mind wandering [106]).

### 4.4. Summary of this Section

Grouping meditations by theoretically meaningful categories resulted in models in which brain patterns (neural imaging patterns or EEG frequencies) were similar within categories but different between categories. An examination of the four-category model brought up two questions. First, two categories were procedural categories (*Focused Attention* and *Open Monitoring*) while the other two were defined by the contents of meditation practice *(Mantra recitation* or *Loving-kindness/compassion)*. These are distinctly different groupings—“procedures used” versus “meditation content”. Second, two of the categories, *Mantra Recitation* and *Loving-kindness/Compassion* could arguably be placed within the *Focused Attention* category since they both involve focusing on a specific target—voluntary verbal-motor production, or a consistent emotional tone—to the exclusion of others. Fox and colleagues discussed this last issue [17].

The three-category model uses EEG patterns associated with three distinct meditation procedures—focused attention, open monitoring, and automatic self- transcending—to objectively assign meditation practices to categories. This creates categories with similar content—procedure related EEG frequencies—and allows meditation practices to move between categories as procedures change with long-term practice, such as from focused concentration to “effortless concentration” [61].

Figure 1a–c display brain areas activated by meditations in each category. The findings of Fox’s meta-analysis of 78 studies were used to identify brain areas active during *Focused Attention* and *Open Monitoring* meditations [8]. Mahone’s fMRI research on Transcendental Meditation practice was used to identify brain areas active during *Automatic Self-Transcending* meditations [49]. This figure includes cortical and sagittal sections of the brain. Brain areas active during *Focused Attention* meditations are colored blue in Figure 1a, are colored yellow during *Open Monitoring* meditations in Figure 1b, and colored red during *Automatic Self-Transcending* meditations in Figure 1c. In addition, the insula was active during *Open Monitoring* meditations, and the pons and cerebellum were *deactivated* during *Automatic Self-Transcending* meditations. The brain’s areas are distinct for each category, engaging different frontal, motor, parietal, and anterior cingulate cortices.

The Brodmann areas are in the background of this figure to help the reader locate the areas that were active. More brain areas may be indicated for *Focused Attention* since this category includes brain areas during Mantra Recitation and Compassion meditation. They were included since they are both characterized by gamma EEG.

In addition, these three categories were marked by distinct EEG patterns. *Focused Attention* meditations were marked by posterior gamma. *Open Monitoring* meditations were marked by frontal midline theta and posterior alpha2. *Automatic Self-Transcending* meditations were marked by frontal alpha1.

## 5. Recommended Grouping Strategy

Grouping meditations by theoretically meaningful categories appears to be the best strategy to define the neurobiology of meditation. This strategy controls for the fatal flaw of combining, into a single average, meditation practices involving outward directed attention to objects or the senses, and inner-directed attention to cognitive and affective processes or nondual states (1st strategy). These different procedures would necessarily involve different brain patterns, and, when averaged together, would define a brain pattern unlike any meditation practice. 

Two examples of combining meditations into meaningful categories were discussed. The four-category model of meditation created by descriptions of the procedures [17] yielded distinct neural patterns within each category with few similarities between categories. The three-category model defined by EEG frequencies associated with meditation procedures yielded distinct EEG patterns within each category with few similarities between categories. In addition, using EEG frequencies to assign meditations to categories allows practices to “move” into different categories as subjects’ meditation experiences change over time, which would be associated with different brain patterns.

## Figures and Tables

**Figure 1 medicina-56-00712-f001:**
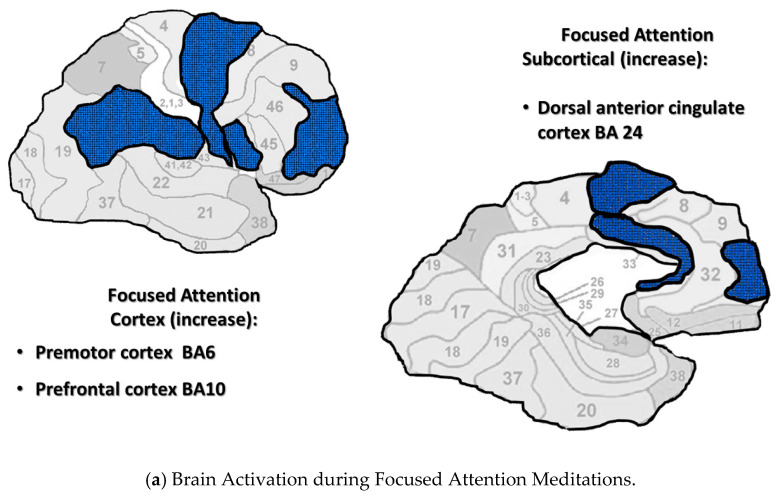
(**a**–**c**). Neural imaging research shows distinct cortical and subcortical activation/ deactivation patterns in three categories of meditation. Cortical and subcortical areas active during *Focused Attention* meditations (**a**), *Open Monitoring* meditations (**b**), and *Automatic Self-Transcending* meditations (**c**).

**Table 1 medicina-56-00712-t001:** Sample of research combining neural images recorded in different meditation traditions in the final averages.

Reference	Design	Result
Luders [5]	22 meditation and 22 age-matched controls. Meditations included Zazen, Shamatha, Vipassana, and Others: 63% used deep concentration, 36% control of breath, 32% visualization, 32% attention to external and internal stimuli, 14% withdrawal of sensory perceptions and 18% letting go of thoughts.	Total grey matter volumes were similar in both groups (meditator and control). There were local differences. Meditators had thicker: Right orbito-frontal cortex; Right thalamus; Left inferior temporal; Right hippocampus.
Luders [6]	27 meditation and 27 age-matched controls. Average age: 52 years.	Fractional anisotropy (FA) (integrity of white matter) was higher in meditators compared to controls within major projection pathways, commissural pathways, and association pathways.
	55% practiced: Shamatha, Vipassana, or Zazen.	
Luders [7]	50 long-term meditators and 50 age-matched controls. Age range: 27–71 years.	Negative correlation of global and local gray matter with age for both meditators and controls. However, the group-by-age interaction was highly significant (*p* = 0.003) with lower grey matter by age for controls.
	Meditations included Zazen, Shamatha, Vipassana.	
Fox [8]	Meta-analysis of 21 neuroimaging studies with N = 300 meditation practitioners: Insight, Zen, Tibetan, Buddhist, Mindfulness Based Stress Reduction, Integrative Mind-Body Training, Soham, Loving-Kindness meditation, and “Various”.	Five regions were thicker in meditators: Meta-awareness (frontopolar cortex); Exteroceptive and interoceptive body awareness (sensory cortices and insula); Memory consolidation and reconsolidation (hippocampus); Self and emotion regulation (anterior and mid cingulate; orbitofrontal cortex); Intra- and interhemispheric communication (superior longitudinal fasciculus; corpus callosum). There was, however, a negative correlation with effect size differences and length of meditation practice.
Lomas [9]	Meta-analysis of 54 studies of EEG from Mindfulness Based Cognitive Training (2), Mindfulness (13), Vipassana (6), mind/body training (2), Zen (13), various (5).	Delta EEG power (5 total): Significantly higher in 1 study, lower in four. Theta EEG power (19 total): Significantly higher in 10 studies, lower in three, and no differences in six others. Posterior Alpha EEG power (20 total): Significantly higher in 13 studies, lower in one, and no differences in six others. Beta EEG power (12 total): Significantly higher in four studies, lower in one, and no differences in seven others. Gamma EEG power (7 total): Significantly higher in three studies, lower in one, and no differences in three others.
Boccio [10]	Meta-analysis with Mindfulness practices (5), Vipassana (12), Kundalini (11), Integrative Body Mind Training (3), MBSR (11), Zen (4).	Areas activated included the medial prefrontal cortex, motor cortex gyrus, anterior cingulate cortex, insula, claustrum, precuneus, parahippocampal gyrus, middle occipital gyrus, inferior parietal lobule, lentiform nucleus and thalamus.
Luders [11]	50 long-term meditators and 50 age-matched controls.	Estimating brain ages. At age fifty, brains of meditators were estimated to be 7.5 years younger than those of controls; For every additional year over fifty, meditators’ brains were estimated to be an additional 1 month and 22 days younger than their chronological age.
	Meditations included Zazen, Shamatha, Vipassana, as in the first study.	
Luders [12]	Meta-analysis of 9 studies: Dzogchen, Loving-kindness meditation, Sahaja yoga meditation, Shamatha, Vipassana, Zen.	Long term meditators (10 years or more) vs. controls had: Higher whole brain grey matter; Thicker cortices overall; High white matter integrity—Higher fractional anisotropy.
Van Aalst [13]	Meta-analysis of 34 studies of Yoga practice: Ashtanga, Iyengar, Vinyasa, Kripalu, Kundalini, Nidra, Sahaja, Sivananda and Hatha yoga.	Yoga practitioners had increased gray matter volume in the insula and hippocampus and increased activation of prefrontal cortical regions. There was, however, high variability in the neuroimaging findings

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
