# Peer review of "On the Neurobiology of Meditation: Comparison of Three Organizing Strategies to Investigate Brain Patterns during Meditation Practice"

_medicina, 2020, doi:10.3390/medicina56120712_

Round 1

Reviewer 1 Report

The author has submitted a manuscript entitled "Neurobiology of Meditation." The manuscript is very difficult to follow.

The ABSTRACT only refers to three strategies of following mediation practices, but the manuscript presents much more information. Therefore, the abstract is insufficient. In the abstract, the author describes the second strategy as "difficult to see the forest for the trees;" this statement is vague and provides no real information. Also, the manuscript seems to be a review article - this point should be included in the abstract.

The INTRODUCTION also refers to those three strategies. No aim or objective is provided, so the reader has no idea what is to follow. The reader can not tell if this manuscript is to be a review article or a research article. If a review article, a brief description of what topics are to be discussed would help greatly organize the manuscript.

 In the section entitled "AVERAGING NEURAL IMAGES...," the author includes a paragraph on search techniques. If this paragraph is intended to be the METHODS section of a systematic review, it should be labeled as METHODS just after the introduction. More information is required about how the review was accomplished. For example, which articles were included and why, and which were discarded.  In not intended as METHODS, then it can be removed.

Multiple sections are labeled as SUMMARY. This organization pattern is very confusing.  If the SUMMARY is referring to the preceding section, then including it as a paragraph in the preceding section with a section label, but beginning the paragraph with "To summarize, ..." may make the manuscript more readable.

Some analysis of the information provided would be helpful. For instance, is one strategy superior to others, and why?

Author Response

 I have indicated the page numbers in this document indicating where I responded to the comment, and I have colored with blue font the new wording in the text. My responses to the reviewer’s comments below are indented. 

Reviewer 1

The ABSTRACT only refers to three strategies of following meditation practices, but the manuscript presents much more information. Therefore, the abstract is insufficient.

The abstract is clarified. It brings out three strategies to explore meditation practices. 1) combining meditations with different practices together in the same study, 2) describing each meditation practice individually and 3) creating apriori categories and comparing meditations within those categories. It discusses three examples of apriori categories that have been used to group meditation studies. Pg 2

In the abstract, the author describes the second strategy as "difficult to see the forest for the trees;" this statement is vague and provides no real information.

Clarified this point by stating the mixing of individual differences within each meditation with common patterns that may exist across meditations practices. Pg 2

Also, the manuscript seems to be a review article - this point should be included in the abstract.

This is not a systematic review or research.  Rather it is a theoretical paper that presents three heuristics to organize research.  I bring this point out in the abstract and the beginning of the introduction pg 4

The INTRODUCTION also refers to those three strategies. No aim or objective is provided, so the reader has no idea what is to follow. The reader can not tell if this manuscript is to be a review article or a research article. If a review article, a brief description of what topics are to be discussed would help greatly organize the manuscript.

I clarified this in the Introduction.  This paper is not an exhaustive review of all papers on meditation practices.  Rather, it compares the brain patterns emerging from these three different methods to group the data. This analysis is to alert researchers to the impact of different organizing strategies to understand the nature and application of meditation practices pg 4

 In the section entitled "AVERAGING NEURAL IMAGES...," the author includes a paragraph on search techniques. If this paragraph is intended to be the METHODS section of a systematic review, it should be labeled as METHODS just after the introduction. More information is required about how the review was accomplished. For example, which articles were included and why, and which were discarded.  In not intended as METHODS, then it can be removed.

This is not a systematic review but a theoretical comparison of three heuristics to organize research.  I removed this paragraph

Multiple sections are labeled as SUMMARY. This organization pattern is very confusing.  If the SUMMARY is referring to the preceding section, then including it as a paragraph in the preceding section with a section label, but beginning the paragraph with "To summarize, ..." may make the manuscript more readable.

Yes, each “summary” is for that section.  I followed your suggestion and removed the heading, and begin the paragraph “To summarize…”  Thank you

Some analysis of the information provided would be helpful. For instance, is one strategy superior to others, and why?

Added a clearer analysis at the end entitled: Comparison of Grouping Strategies pg 23-4

Reviewer 2 Report

This article is interesting but it also contains a number of problematic flaws. In addition, despite its title, it presents too narrow of a review of available literature on the neurobiology of meditation. Granted that literature is quite extensive, but several summarizing articles have not received attention here. While there is limited discussion of some recent meta-analyses, others on meditation and fMRI have not been addressed in terms of how different types of meditation practices appear to have impact on either different areas of the brain, or more significantly on  different combinations of activations and deactivations in the brain, e.g., Falcone & Jerram, 2018; the larger meta-analysis (78 vs 21 studies) by Fox, Dixon, et al., 2016; Kim, Cunningham, Kirby, 2020; but also see Shen, Zhou, Chen, Castellanos, & Yan, C. G., 2020, which adds support for some of the findings reported here.

One problem is that EEG studies are often not conclusive because of the varying meditation styles and practitioners assessed, as noted in this article. However, neuroimaging studies are increasingly finding similar patterns during meditation in long-term meditators; but with patterns differing across unique meditation traditions (see e.g., Brandmeyer, Delorme, & Wahbeh, 2019). This issue should receive attention to create a more balanced view. That the author has competence in neuroscience and meditation is not in doubt, but sometimes there seems to be an overreliance on specific authors, e.g., Luders. In addition, results from longitudinal versus novice meditator research could be better addressed.

There are also some assumptions that may go too far in terms of methodology. For example, the attempt to group meditation practices by “level of effort” is problematic as presented. Certain (TM) biases are perhaps also indicated. There is virtually no direct information on so-called “Vedic” meditation practices, although addressing Hindu and Buddhist meditation differences does make sense.

The author’s analysis of the Yoga Sutra is very disturbing. First, citta does not mean “mind” as the Sanskrit term for mind is “manas”; citta refers to “thought”. In addition, nirodha is well understood to imply “cessation” (as noted in the article), an ending or stopping (of thought); and if it pleases the author this could be discussed as a form of “control” of mental activity. But let’s be clear here, this procedure also includes: “manas”, “buddhi”, and “ahamkara” (as well as the physical sense faculties - indirya). Many contemporary yoga and meditation traditions like to point to the Yoga Sutra as their source text- but this is not usually accurate for a large number of reasons, and this text or any of its creative mistranslations are not useful as a basis for organizing modern meditation procedures. In addition, the ashtanga (“8 limbs”) of Patanjali’s yoga discussions are not all equivalent as means for achieving kaivalya, anymore so then they are in the historical/medieval Hatha Yoga traditions of South Asia.

Finally, neuroscience is still debating the “default mode network” (DMN) and if you want to focus here you should also likely discuss the so-called “salience network” (SN) as well. The paragraph that includes the discussion about the “inherent pleasure” of yoga is unfounded and contains several disturbing claims.

Overall, this article does provide some interesting information, especially in regard to EEG measurements recorded during different meditation practices, but it should be more straightforward about its potential contribution. With revision, a title like “EEG analysis of meditation practices” would be more accurate. In this reviewer’s view, this article needs moderate revision.

Author Response

 I have indicated the page numbers in this document indicating where I responded to the comment, and I have colored with blue font the new wording in the text. My responses to the reviewer’s comments below are indented. 

Reviewer 2

This article is interesting but it also contains a number of problematic flaws. In addition, despite its title, it presents too narrow of a review of available literature on the neurobiology of meditation. Granted that literature is quite extensive, but several summarizing articles have not received attention here. While there is limited discussion of some recent meta-analyses, others on meditation and fMRI have not been addressed in terms of how different types of meditation practices appear to have impact on either different areas of the brain, or more significantly on  different combinations of activations and deactivations in the brain, e.g.,

 Thank you for these references.

    • Falcone & Jerram, 2018;Is added to individual section on mindfulness pg 12
    • Fox, Dixon, et al., 2016; is it’s own section on the 3rd strategy pg 17 - 18
    • Kim, Cunningham, Kirby, 2020; added to individual section on compassion pg 12
    • Shen, Zhou, Chen, Castellanos, & Yan, C. G., 2020, is added to Default Mode Network section.pg 22
    • Kemmer in Zen section pg 12

One problem is that EEG studies are often not conclusive because of the varying meditation styles and practitioners assessed, as noted in this article. However, neuroimaging studies are increasingly finding similar patterns during meditation in long-term meditators; but with patterns differing across unique meditation traditions (see e.g.,

 Added Brandmeyer, Delorme, & Wahbeh, 2019 when discussed focused attention meditations can lead to “effortless concentration.” In a new section Automaticity: Why Zen Buddhism and Qigong Might be in this Category Pg 20 - 21

This issue should receive attention to create a more balanced view. That the author has competence in neuroscience and meditation is not in doubt, but sometimes there seems to be an overreliance on specific authors, e.g., Luders. In addition, results from longitudinal versus novice meditator research could be better addressed.

I have reduced Luder references and added others.  I have endeavored to make it more balanced throughout

There are also some assumptions that may go too far in terms of methodology. For example, the attempt to group meditation practices by “level of effort” is problematic as presented. Certain (TM) biases are perhaps also indicated.

Patterned this discussion after Fox 2016, using description of procedures to group practices to show differences in cognitive control of different practices  pg. 16

There is virtually no direct information on so-called “Vedic” meditation practices, although addressing Hindu and Buddhist meditation differences does make sense.

Changed to Buddhist and Indian.  Hinduism is a religion, while Buddhism is not.  Buddhism, as I understand, was begun in Nepal.  So Indian and Buddhism seems a better dichotomy here

The author’s analysis of the Yoga Sutra is very disturbing. First, citta does not mean “mind” as the Sanskrit term for mind is “manas”; citta refers to “thought”. In addition, nirodha is well understood to imply “cessation” (as noted in the article), an ending or stopping (of thought); and if it pleases the author this could be discussed as a form of “control” of mental activity. But let’s be clear here, this procedure also includes: “manas”, “buddhi”, and “ahamkara” (as well as the physical sense faculties - indirya). Many contemporary yoga and meditation traditions like to point to the Yoga Sutra as their source text- but this is not usually accurate for a large number of reasons, and this text or any of its creative mistranslations are not useful as a basis for organizing modern meditation procedures. In addition, the ashtanga (“8 limbs”) of Patanjali’s yoga discussions are not all equivalent as means for achieving kaivalya, anymore so then they are in the historical/medieval Hatha Yoga traditions of South Asia.

I apologize.  I have deleted this paragraph and this line of thinking

Finally, neuroscience is still debating the “default mode network” (DMN) and if you want to focus here you should also likely discuss the so-called “salience network” (SN) as well. The paragraph that includes the discussion about the “inherent pleasure” of yoga is unfounded and contains several disturbing claims.

I have re-written the default mode network discussion and expanded the discussion and discussed negative and positive mind wandering.  This is a debated topic and I tried to give a fuller picture pg 21 – 23.

Overall, this article does provide some interesting information, especially in regard to EEG measurements recorded during different meditation practices, but it should be more straightforward about its potential contribution. With revision, a title like “EEG analysis of meditation practices” would be more accurate. In this reviewer’s view, this article needs moderate revision.

I re-titled the article to more clearly illustrates its contents.  It is now “Comparison of Three Organizing Strategies to Investigate Brain Patterns during Meditation Practice.”  I used this title since I present both neural imaging and EEG patterns to explore three organizing strategies

Reviewer 3 Report

In this article, the author discussed strengths and limitations of three strategies to study meditation. They nicely summarised the neurobiology of different practices and concluded that the organising strategy based on effort used during mediation provides a better assessment of different meditation practices and hence could provide more reliable results.

The article is very well written, and the topic is timely.

There are a few points that could potentially improve the article.

  1. Line 35: “Signal averaging is an established procedure in EEG …” does not seem strictly related to the “averaging-organizing principle to meditation practices”. Perhaps, one could provide a better example on averaging brain activity over different cognitive tasks (where different brain areas are activated) which would be more relevant to the “averaging-organizing principle to meditation practices”.
  2. Line 94: The abbreviation “MRI” needs to be defined.
  3. Line 213: “… are characterised by gamma EEG”. Indeed, it would be important to specify (even roughly) the brain areas of the gamma activity in order to understand its spatial specificity (e.g. van Lutterveld et al., 2017, NeuroImage, 151: 117–127). This will also help to link EEG and fMRI findings.
  4. Line 222: “Alpha power is usually considered a sign of cortical idling”; however, the alpha oscillations also demonstrate an active (inhibitory) role in attention and memory tasks (e.g. Jensen and Mazaheri, 2010, Front Hum Neurosci, 4: 186).
  5. Line 229: “EEG frequencies are one objective marker of effort during meditation practices”. Here the word “effort” requires a more concrete definition. For instance, in a working memory task the effort (to memorise the items) is often proportional to the task difficulty (e.g. 1 vs. 4 items) and it can be quantified via behavioural performance (e.g. error rate, response time, etc.). However, in mediation the effort cannot be easily quantified and thus, it does seem subjective. Because the effort-based organisation strategy is one of the key concepts of this article, the author needs to provide a solid ground on how to assess effort (behaviourally) in different meditation practices.
  6. Line 253: “minimal effort”, please provide a more detailed explanation.
  7. Line 254: “high levels of activation of the default mode network”; maybe, just higher activation (?)

Author Response

I have indicated the page numbers in this document indicating where I responded to the comment, and I have colored with blue font the new wording in the text. My responses below are indented to the reviewers 

Reviewer 3

In this article, the author discussed strengths and limitations of three strategies to study meditation. They nicely summarized the neurobiology of different practices and concluded that the organizing strategy based on effort used during meditation provides a better assessment of different meditation practices and hence could provide more reliable results.The article is very well written, and the topic is timely.

There are a few points that could potentially improve the article.

  1. Line 35: “Signal averaging is an established procedure in EEG …” does not seem strictly related to the “averaging-organizing principle to meditation practices”. Perhaps, one could provide a better example on averaging brain activity over different cognitive tasks (where different brain areas are activated) which would be more relevant to the “averaging-organizing principle to meditation practices”.

I changed the thrust of this first strategy from “averaging” to finding a “common underlying pattern” and then gave intelligence research to find ‘g’  and signal averaging. Pg 4-5

  1. Line 94: The abbreviation “MRI” needs to be defined.

Done

  1. Line 213: “… are characterised by gamma EEG”. Indeed, it would be important to specify (even roughly) the brain areas of the gamma activity in order to understand its spatial specificity

Added this reference to discussion of gamma when discussing focused Attention meditations  van Lutterveld et al., 2017, NeuroImage, 151: 117127). pg 19.

  1. Line 222: “Alpha power is usually considered a sign of cortical idling”; however, the alpha oscillations also demonstrate an active (inhibitory) role in attention and memory tasks

Added Jensen and Mazaheri, 2010, Front Hum Neurosci, 4: 186 and Palva and Palva in discussing Automatic Self-Transcending  .pg 20

  1. Line 229: “EEG frequencies are one objective marker of effort during meditation practices”. Here the word “effort” requires a more concrete definition. For instance, in a working memory task the effort (to memorise the items) is often proportional to the task difficulty (e.g. 1 vs. 4 items) and it can be quantified via behavioural performance (e.g. error rate, response time, etc.). However, in mediation the effort cannot be easily quantified and thus, it does seem subjective. Because the effort-based organisation strategy is one of the key concepts of this article, the author needs to provide a solid ground on how to assess effort (behaviourally) in different meditation practices.

 Since people are  meditating it is difficult to behaviorally define “effort” or “cognitive control.”  .  I used descriptions of procedures as Fox 2016 did to bring out the range of meditation practices and substantiated them with EEG frequency bands associated with that level of cognitive control. Pg 19-20

  1. Line 253: “minimal effort”, please provide a more detailed explanation.

Used descriptions of practices to make this point. And fuller discussion of “effortless awareness”   Pg 20 - 21

  1. Line 254: “high levels of activation of the default mode network”; maybe, just higher activation (?)

Done

Round 2

Reviewer 1 Report

The authors have satisfactorily addressed the reviewer's comments. The manuscript is now much easier to follow.

A brief conclusion of three or four sentences would nicely wrap up the manuscript.

Author Response

Thanks